Multi-granularity adaptive extractive document summarization with heterogeneous graph neural networks

http://orcid.org/0000-0002-8229-4799 Su Wu
Jiang Jin 15674352@qq.com
Huang Kaihui
School of Automation and Electronic Information, Xiangtan University , Xiangtan, Hunan Province , China
Kong Xiangjie
Electronic publication date: 2023 Dec 13
Publication date: 2023
Volume: 9
Electronic Location ID: e1737
Received 2023 May 4; Accepted 2023 Nov 14
Copyright: © 2023 Su et al.
Copyright year: 2023
Copyright holder: Su et al.
License: This is an open access article distributed under the terms of the Creative Commons Attribution License, which permits unrestricted use, distribution, reproduction and adaptation in any medium and for any purpose provided that it is properly attributed. For attribution, the original author(s), title, publication source (PeerJ Computer Science) and either DOI or URL of the article must be cited.
License URL: https://creativecommons.org/licenses/by/4.0/

Keywords: Extractive summarization, Graph neural networks, Adaptive, Graph attention network

Funding: Hunan Province Natural Science Foundation of China under Grant 2021JJ30671 This work was supported by the Hunan Province Natural Science Foundation of China under Grant (2021JJ30671). The funders had no role in study design, data collection and analysis, decision to publish, or preparation of the manuscript.

==============================
The crucial aspect of extractive document summarization lies in understanding the interrelations between sentences. Documents inherently comprise a multitude of sentences, and sentence-level models frequently fail to consider the relationships between distantly-placed sentences, resulting in the omission of significant information in the summary. Moreover, information within documents tends to be distributed sparsely, challenging the efficacy of sentence-level models. In the realm of heterogeneous graph neural networks, it has been observed that semantic nodes with varying levels of granularity encapsulate distinct semantic connections. Initially, the incorporation of edge features into the computation of dynamic graph attention networks is performed to account for node relationships. Subsequently, given the multiplicity of topics in a document or a set of documents, a topic model is employed to extract topic-specific features and the probability distribution linking these topics with sentence nodes. Last but not least, the model defines nodes with different levels of granularity—ranging from documents and topics to sentences—and these various nodes necessitate different propagation widths and depths for capturing intricate relationships in the information being disseminated. Adaptive measures are taken to learn the importance and correlation between nodes of different granularities in terms of both width and depth. Experimental evidence from two benchmark datasets highlights the superior performance of the proposed model, as assessed by ROUGE metrics, in comparison to existing approaches, even in the absence of pre-trained language models. Additionally, an ablation study confirms the positive impact of each individual module on the model's ROUGE scores.

Introduction

The critical component of extractive summarization involves an in-depth analysis of the source text to produce condensed, semantically clear sentences that serve as a summary of the document. Prevailing models often utilize an encoder-decoder framework based on attention mechanisms and yield satisfactory outcomes in the realm of short text summarization. Recurrent neural networks (RNNs) (Cheng & Lapata, 2016; Nallapati, Zhai & Zhou, 2017; Zhou et al., 2018) commonly serve the purpose of elucidating cross-sentence relationships. Nonetheless, these RNN-based frameworks often fall short in addressing long-distance dependencies, particularly when handling extensive or multi-document texts. An alternative avenue worth exploring is the utilization of graph structures for modeling inter-sentence connections. Yet, the effective design of these graph structures for automating document summarization remains an area warranting further research.

Documents inherently comprise a multitude of sentences, and sentence-level models frequently fail to consider the relationships between distantly-placed sentences, resulting in the omission of significant information in the summary. Moreover, information within documents tends to be distributed sparsely, challenging the efficacy of sentence-level models. To address these shortcomings, a fused multi-granularity node graph model is proposed. This model integrates a topic model to isolate keyword features, and leverages both TF-IDF and topic probabilities for the generation of edge features, thus strengthening the interconnections among nodes of varying granularities. Through iterative node information propagation, links are established between sentences, topics, and words, facilitating enhanced information transfer to both proximal and distant nodes. Additionally, an adaptive mechanism for adjusting the depth and breadth of the graph is incorporated, progressively improving the model’s capacity to handle long-distance dependencies.

Semantic units in the model are composed of words in conjunction with topics. Each sentence is interconnected with the words and topics it encompasses, yet subjects and words lack direct edges between them. Edge features are incorporated into the calculations of the graph attention layer, facilitating information transfer between nodes. Given the varying requirements for propagation width and depth across nodes of different granularities, adaptability in both dimensions is introduced to enhance the model’s performance. The merits of the constructed multi-granularity adaptive heterogeneous graph can be outlined as Xiangtan University, Xiangtan City, Hunan Province follows: (a) Sentence nodes engage in interactions via word nodes and topic nodes, compensating for explicit word information overlaps. (b) The incorporation of a topic model enriches the model with high-level semantic nodes, thereby considering the semantic associations behind words to bolster the model’s classification capabilities. (c) Both topic and word nodes have the opportunity to update their node representations through information aggregation from sentence nodes. (d) The calculation of attention coefficients in the graph attention layer is subject to relative change, with edge features participating in the computation, allowing for dynamic adjustments in the graph attention layer to more effectively consider inter-node relationships. (e) Through the mechanism of adaptive width and depth, information at varying levels of granularity is efficiently accounted for.

The contributions of this work can be summarized as follows:

(1) The inclusion of topic nodes allows the model to adeptly handle texts with similar content but fewer identical words, thereby enhancing its classification ability.

(2) In the graph attention layer, the challenge of relatively static attention coefficients is addressed by adding edge features to the calculation, thereby making effective use of the semantic content within these features.

(3) Adaptability in terms of node width and depth enables the learning of importance and correlation between nodes with different granularities.

Related work

Extractive document summarization

The significance and interrelationships among nodes of varying granularities are ascertained through mechanisms of adaptive width and depth. The majority of related studies concentrate on utilizing the encoder-decoder framework, often employing recurrent neural networks for this purpose (Cheng & Lapata, 2016; Nallapati, Zhai & Zhou, 2017; Zhou et al., 2018). Additionally, the adoption of pre-trained language models has gained traction for enabling contextual word representations in summarization tasks (Zhou et al., 2018; Liu & Lapata, 2019b; Zhong et al., 2019b).

Graph structures serve as another intuitive approach for extractive summarization, offering an enhanced capacity to exploit statistical or linguistic correlations between sentences. Initial efforts in this domain were centered on document graphs assembled based on content similarity among sentences, as exemplified by algorithms like LexRank (Erkan & Radev, 2004) and TextRank (Mihalcea & Tarau, 2004). More recent studies have ventured into the dissection of documents into nodes at diversified levels of granularity, employing heterogeneous graph neural networks for this objective.

Heterogeneous graph for NLP

Graph neural networks and their associated learning methodologies (Gilmer et al., 2017; Veličković et al., 2017) were initially conceived for homogeneous graphs, wherein all nodes share a uniform type. However, real-world applications often involve graphs with diverse types of nodes, or heterogeneous graphs. Recent studies have initiated exploratory efforts to model these complex structures (Shi et al., 2016). For instance, researchers have introduced heterogeneous graph neural networks to encode documents, entities, and candidates simultaneously for tasks like multihop reading comprehension (Tu et al., 2019). Others have focused on semi-supervised short text classification, employing a topic-entity heterogeneous neural graph (Linmei et al., 2019). In the realm of automatic document summarization, various approaches have been proposed, such as designing a heterogeneous graph comprising topic, word, and sentence nodes and employing a Markov chain model for iterative node updates (Wei, 2012). Algorithms like HeteroRank extend existing models (Wang, Chang & Huang, 2019) like TextRank by incorporating additional elements like keywords and sentences. Other models like HDSG Wang et al. (2020) employ words, sentences, and even paragraphs to form graph structures. The HEROS (Jia et al., 2020) model applies graph-based techniques to long-text fields, taking into account input article discourse. These methodologies primarily showcase two categories of variations: those pertaining to graph construction and those modifying the prior values of graph nodes. As for graph construction, the principal alterations relate to vertex granularity and edge weights. Some approaches operate under the assumption that significant sentences are composed of important words, employing a hybrid of graph models and sorting algorithms to rank both words and sentences (Fang et al., 2017). Other methods introduce more complex heterogeneous graphs that include not just words and sentences as nodes, but also integrate topic information. Yet another set of studies, such as those employing ExpandRank, utilize information from adjacent documents to enhance the sentence graph of a target document (Yang et al., 2018).The current approach maximizes the utility of both local document information and external data for evaluating the significance of sentences (Wan & Xiao, 2010). The method also incorporates topic information to bolster summarization efficacy. A distinguishing feature of this methodology is its incorporation of adaptive depth and breadth, facilitating the handling of long-distance dependencies across multiple layers. Additionally, the model adjusts the level of connectivity for each graph neural network (GNN) layer according to the local neighborhood size of individual nodes. Moreover, topic probability distribution values are employed as edge features between topic nodes and sentence nodes, thereby enabling the latter to aggregate more pertinent topics. Separate research has introduced innovative graph models for text summarization (Ferreira et al., 2013), constructing graphs based on a four-dimensional framework: similarity, semantic similarity, co-citations, and textual information. With regard to modifications in prior values, diverse strategies exist for assessing the preliminary importance of nodes within the graph. For example, a biased TextRank algorithm has been proposed to extract focused content by modifying the random restart probability based on node relevance to the targeted task (Kazemi, Pérez-Rosas & Mihalcea, 2020). Such a modification skews the algorithm toward selecting nodes of higher relevance. This particular methodology begins by scoring sentences using a pre-trained model, employing these scores to establish the preliminary importance of the sentences within the graph.

In recent years, the advent of distributed representations in natural language processing (Mikolov et al., 2013; Devlin et al., 2018) has led to a proliferation of deep learning-based techniques for automatic summarization (An et al., 2021; Liu, 2019). These methods have made strides in enhancing the efficacy of automatic document summarization, yet the task of managing long-distance dependencies continues to pose a significant challenge, and the training procedures are often resource-intensive. Notwithstanding, textual semantic information can be efficiently mapped as nodes within a graph-based framework. This configuration augments the model’s information flow and elevates the coherence of the generated summaries. The architecture also enables the efficient encoding of information across multiple levels of granularity. As a result, several methodologies have integrated multi-granular information with graph models to amplify text summarization effectiveness. In the present study, a sentence-topic-word structure has been established, proving instrumental in addressing cross-sentence dependencies for both single-document and multi-document summarization tasks (Chen, 2023; Mao et al., 2021; Shafiq et al., 2023). Moreover, an approach employing adaptive breadth and depth has been implemented to update nodes of varied granularity, thereby facilitating a more effective aggregation of node-specific information.

Methodology

In the realm of extractive document summarization, the interplay among textual elements exerts a significant influence on the identification of essential content. A multi-granularity adaptive extractive document summarization framework based on a heterogeneous graph neural network is introduced in this research. This framework comprises three primary components: a graph initializer, adaptive heterogeneous graph layers, and a sentence selector. As illustrated in Fig. 1, the model’s architecture is presented comprehensively. Initially, the graph initializer generates nodes and edges at three levels of granularity—word, sentence, and topic—and performs encoding tasks for these entities. Subsequently, the adaptive heterogeneous graph layers facilitate the message passing between nodes of varied granularity, leading to the updating of their respective features. Ultimately, the characteristics gleaned from the updated sentence nodes are utilized for label prediction.

Figure 1 Overall framework.

The framework consists of three main modules: graph initializer, heterogeneous layer and sentence selector. W represents the word node, S and T represent the sentence node and topic node respectively, and the orange solid line is the edge feature (TF-IDF, probability distribution) between nodes.

Modeling heterogeneous graphs

A heterogeneous graph G is defined as G={V,E}, where V represents nodes of various classes and E represents edges between nodes. Detailed definition V=Vw∪Vs∪Vt, E=Ews∪Ewt, where Vw={w1,...,wm} denotes m unique words of the document, Vs={s1,...,sn} corresponds to the n sentences in the document and Vt={t1,...,tn} denotes j topics in the article. E is a real-value edge weight matrix and Ewsij≠0 (i∈{1,...,m},j∈{1,...,n}) indicates the j-th sentence contains the i-th word, Estuv≠0 (u∈{1,...,n},v∈{1,...,g}) indicates the u-th sentence contains the v-th topics.

Graph initializers

In the graph initialization phase, vector representations for each node category are generated through the utilization of GloVe embeddings. GloVe (Pennington, Socher & Manning, 2014) serves as a word embedding technique designed for the conversion of a vocabulary into a vector space, capitalizing on co-occurrence data from extensive text corpora. This results in high-quality vector representations that enrich the model’s comprehension of textual semantics. When compared with Transformer-based language models such as BERT (Devlin et al., 2018) and GPT (Radford & Narasimhan, 2018), GloVe embeddings require lower computational costs and offer quicker query speeds, thereby exerting minimal impact on the training pace of the model. Following this, features corresponding to sentences, topics, and words are extracted from the document. The architecture of the Transformer (Vaswani et al., 2017) has been shown to excel in the extraction and analysis of document features, employing self-attention mechanisms that permit the model to concentrate on various positions within the input sequence in a concurrent manner. This allows for effective capturing of long-range dependencies. Nevertheless, the Transformer model necessitates the retention of a complete self-attention matrix, which can lead to memory constraints when lengthy sequences are processed. Moreover, the performance of the Transformer is often contingent upon expansive pretraining datasets, demanding considerable volumes of textual data for its initial training. Given these considerations surrounding training time and computational resource needs, this study opts for CNNs (LeCun et al., 1998) furnished with diverse kernel sizes for the extraction of local features. Additionally, bidirectional long short-term memory (BiLSTM) (Graves & Graves, 2012) networks are employed to extract more global features during the feature extraction phase.

Specifically, let Xw∈Rm∗ds, Xs∈Rn∗ds and Xt∈Rj∗dt represent the input feature matrices corresponding to word nodes, sentence nodes, and topic nodes, respectively. Here, dw signifies the dimension of the word vector, ds denotes the dimension of the sentence vector, and dt indicates the dimension of the topic vector. For the extraction of topic and sentence features, CNNs equipped with various kernel sizes are utilized to obtain local n-gram features, represented as Lj. Subsequently, a BiLSTM is deployed to capture global features, denoted as Gj. The resultant sentence node features and topic node features are formulated as Xsj=[Lj;Gj] and Xti=[Li;Gi], through the concatenation of CNN local features and BiLSTM global features.

CNNs furnished with different kernel sizes have the capability to extract features at multiple scales, enabling the model to capture the document’s content in a comprehensive manner. Moreover, BiLSTM maps each word in the document to its corresponding representation within the sentence, thereby incorporating the contextual nuances of the vocabulary. This contributes to an enhanced understanding of the contextual relationships among words within the document.

As for topic nodes, which serve as more complex semantic units, probability distribution values are obtained through the latent Dirichlet allocation (LDA) (Blei, Ng & Jordan, 2003) topic model and are incorporated into the edge weights between the topic and sentence nodes. This process ensures that edge features encapsulate crucial relationships between the topic and sentence nodes. Likewise, TF-IDF values are infused into the edge weights connecting word nodes and sentence nodes. Term frequency (TF) refers to the occurrence frequency of a word within a document, while inverse document frequency (IDF) represents the inverse function of a word’s outdegree.

Graph neural network layer

Graph attention layer

After the initialization of various node features and edge features in heterogeneous graph G,Considering the relatively constant attention coefficient of a set of keys for different queries in graph attention network (GAT) (Veličković et al., 2017), an improved graph attention network(GATv2) (Brody, Alon & Yahav, 2021) is used to update the representation of nodes. The dynamic attention coefficient used to calculate the two nodes in GATv2 changes relatively but the influence of edge features on the calculation of the attention coefficient of two nodes of different granularity is not considered, so the edge feature eij is added to the calculation. Set hi∈Rdh,i∈{1,...,(m+n)} as the hidden state of the input nodes. GATv2 with added edge features is as follows:

Zij=WaLeakyReLU([Wqhi+Wkhj+Wgeij])

αij=exp(zij)Σl∈Niexp(zil).

ui=σ(∑j∈Ni∝ijWuhj).

where Wa, Wq, Wk, Wu and Wg are trainable weights, and ∝ij is the attention weight between hi and hj. Multi-head attention can be expressed as follows:

ui=∥k=1Kσ(∑j∈Ni∝ijkWkhj).

Adaptive layer

Since there are nodes of different thickness and fine granularity defined in the graph structure, nodes of different granularity need different propagation widths and depths to capture the complex relationship between propagated information. Therefore, it is necessary to learn the importance and correlation between nodes of different granularity by means of node adaptive width and depth. For adaptive width, the feature vectors of neighbor nodes are weighted and summed by improved GATv2 to aggregate 1-hop features. For the adaptive depth, the forgetting and saving between different hops can be controlled by fusing an LSTM module. In traditional GAT, information between different hop can be implicitly fused through multi-layer GAT. However, it is difficult for this fusion method to ensure that only features of feature regions are fused and features of other regions are discarded, while LSTM can automatically forget some links to ensure feature fusion. The adaptive width is completed by GATv2, and the adaptive depth layer is designed as follows:

Ii=σ(Wi(t)Tui).

fi=σ(Wf(t)Tui).

oi=σ(Wo(t)Tui).

C~=tan⁡h(Wi(t)Tui).

Ci(t+1)=fi⊙Cit+Ii⊙C~.

hi(t+1)=oi⊙tan⁡h(Ci(t+1)).

The gating unit Ii with sigmoid output is used to extract new useful signals from C~, and the two are added to the memory after dot multiplication Ii⊙C~. The gating unit fi is used to filter useless signals in the old memory, and a new neighborhood is given. Therefore, the memory of each node can be filtered as Ci(t+1).

Finally, the embedding value hi(t+1) of node i in (t+1)−t layer can be output by using the gating unit oi and the latest memory Ci(t+1). The overall adaptive width and depth layers are shown in Fig. 2.

Figure 2 Adaptive model width and depth, the width of the adaptive use GATv2, depth of adaptive use LSTM control between different hop forgotten and preservation, output node i in (t+1)−t layer embedded value hi(t+1).

Iterative updating

The topic, word, and sentence nodes in the diagram structure need to pass messages to each other for updates, so we design message propagation as Fig. 3. Specifically, after the graph attention layer has processed the data, the data is then entered into a position-wise feed-forward (FFN) layer consisting of two linear transformations, like Transformer (Vaswani et al., 2017).

Figure 3 Detailed update process of each node in the heterogeneous layer.

The green and blue nodes represent the nodes participating in the message transfer. The orange edges indicate the direction of this information transfer. Firstly, the sentence node S1 uses the word nodes W1 and W2 to aggregate the word-level information in it, and then the topic nodes T1 and T2 are used to aggregate the toy-level information. Finally, the sentence node updates the word nodes and topic nodes in turn.

After initializing the nodes, we use neighbor nodes to update the feature representation of sentence nodes with improved GATv2 and FNN layer and adaptive depth layer. The details are as follows:

Us←w1=GAT(Hs0,Hw0,Hw0).

H^s1=FFN(Depth(Us←w1)+Hs0).

Us←w1=GAT(H^s1,Ht0,Ht0).

Hs1=FFN(Depth(Us←t1)+H^s1).

where Hw1=Hw0=Ws, Hs0=Xs and Hs←w0∈Rm∗dh. depth() is the adaptive depth layer, GAT(Hs0,Hw0,Hw0) indicates that Hs0 is used as the attention query, and Hw0 is used as the key and value.

Then, the updated sentence nodes are brought into the calculation together with the unupdated word nodes or topic nodes to obtain the updated representations of the word nodes and topic nodes, and the updated word nodes and topic nodes are used to further iteratively update the sentence nodes. Denote the process of t iterations as follows:

Ut←st+1=GAT(Htt,Htt,Hst).

Htt+1=FFN(Depth(Ut←st+1)+Htt).

Uw←st+1=GAT(Hwt,Hst,Hst).

Hwt+1=FFN(Depth(Uw←st+1)+Hwt).

Us←tt+1=GAT(Hst,Htt+1,Htt+1).

H^st+1=FFN(Depth(Us←wt+1)+Hst).

Us←wt+1=GAT(H^st+1,Hwt+1,Hwt+1).

Hst+1=FFN(Depth(Us←wt+1)+H^st+1).

As shown in Fig. 3, document-level information in sentences is aggregated into topic nodes and word nodes in order to fuse information effectively. For example, if a topic node has a high probability in many sentences, it is likely to be a key topic for this document. In the sentence nodes, the sentence focus often selects the nodes with more words and key topics as the summary.

Sentence selection

In the final stage, sentence nodes are classified to identify those that contain words of greater importance within the heterogeneous graph. The model undergoes training, targeting minimization of cross-entropy loss. Trigram Blocking is employed for the decoding phase, wherein each sentence is ranked based on its associated score, and sentences featuring overlapping letter combinations with preceding ones are discarded.

Trigram Blocking

Building upon the noteworthy outcomes achieved in the works by Paulus, Xiong & Socher (2017) as well as Liu & Lapata (2019b), Trigram Blocking is implemented for decoding within the model. This specific version of maximum marginal correlation (Carbonell & Goldstein, 1998) is not only potent but also relatively straightforward. In particular, sentences are ranked based on the scores they receive within the model, and those with trigram overlaps with earlier sentences are subsequently eliminated.

Multi-document summary

In multi-document summarization, the interrelationships between document-level nodes play an indispensable role for the model’s ability to discern core topics and prioritize content across multiple documents. Yet, many extant models neglect this hierarchical organization, opting instead for a flat, sequential arrangement of multiple documents (Liu et al., 2018). Alternative approaches attempt to model these document-level relationships through attention-based, fully-connected graphs, or by leveraging similarity and statement relationships (Liu & Lapata, 2019a).

To establish document-level relationships analogously to sentence-level counterparts, the model can be effortlessly adapted for multi-document summarization by incorporating super-nodes (as shown in Fig. 4) for each document. Specifically, the heterogeneous graph is augmented with four distinct types of nodes: Vw∪Vs∪Vd∪Vt, Vd={d1...dl}, l represents the number of documents.

Figure 4 State trajectories of oscillator along one dimension in Example 1.

As illustrated in Fig. 4, word nodes serve as the intermediary between sentence and document nodes, with relationships being established based on content similarity. Topic nodes directly link with sentence nodes according to the document’s importance.

A document node, which is essentially a specialized form of a sentence node but with extended content, associates with its corresponding word node, utilizing the TF-IDF value as the edge weight. The updating procedure for document nodes parallels that of sentence nodes, with the key distinction being in initialization; the document node is initialized via mean-pooling of the sentence node features. During the sentence node selection process in multi-document mode, the representation of each sentence node is concatenated with the document representation, leading to the final score post multi-document aggregation.

Experiment

Model experiments were executed on two disparate datasets, yielding ROUGE evaluation metrics (Lin, 2004; Akhmetov, Mussabayev & Gelbukh, 2022) for both single-document and multi-document summarization tasks. Due to constraints on computational resources, experiments involving pre-trained models were not conducted; instead, the investigations commenced based on dataset specifications.

Datasets

CNN/DailyMail

This dataset (Hermann et al., 2015; Nallapati et al., 2016) serves as the most prevalent benchmark for single-document summarization tasks. The standard partition of the dataset comprises training, validation, and testing examples. Data preprocessing adheres to the methodology set forth by Liu & Lapata (2019b), utilizing the non-anonymized version, as recommended by See, Liu & Manning (2017), to obtain ground-truth labels.

Multi-News

Introduced by Fabbri et al. (2019), the Multi-News dataset is a comprehensive collection designed specifically for multi-document summarization. It encompasses article-summary pairs, each consisting of 2–10 original articles alongside a human-generated summary. During the experiment, the dataset is divided into training, validation, and testing sets, with the input articles truncated to a maximum of 500 tokens.

Evaluation

We used ROUGE (ROUGE-1, ROUGE-2, ROUGE-L) to evaluate our model. The formula is as follows:

ROUGEN=∑s∈ref∑N−gram∈scountmatch(N−gram)∑s∈ref∑N−gram∈scount(N−gram)).

ref represents the reference summaries. countmatch (N−gram) represents the quantity of matches generated by the algorithms in comparison with the reference summaries. Similarly, countmatch (N−gram) indicates the tally of matches that the algorithms produce when aligned with the reference summaries. Count (N−gram) signifies the total number of N−gram present in the reference summaries.

Experimental settings

The vocabulary is restricted to 50,000, with token initialization using 300-dimensional GloVe embeddings. A total of 6,214 word nodes are created, while filtering out stop words and punctuation. The input document is truncated to 50 sentences of maximum length. To mitigate the impact of noisy common words, 10% of the vocabulary with the lowest TF-IDF values across the entire dataset is eliminated. Sentence nodes, phrase nodes, and entity nodes are initialized as ds=dp=dv=128, and the edge feature eij is set to de=50. Each GATv2 layer consists of eight heads, the hidden size is dh=64, and the internal hidden layer size of the FFN layer is 512.

During training, a batch size of 32 is employed, and the Adam optimizer (Fabbri et al., 2019) is utilized with a learning rate of 5e−4. The iteration number t=1 is selected based on performance on the validation set. Regarding the length of abstracts, the CNN/DailyMail dataset incorporates the first three sentences, while the Multi-News dataset includes the first nine sentences, based on the average length of manually generated abstracts.

Models for comparison

Ext-BiLSTM

This extractive summarizer employs a BiLSTM encoder to discern relationships between sentence sequences. For the sake of simplicity, the initialization of sentence nodes for classification is adopted, incorporating a CNN encoder at the word level and a two-layer BiLSTM at the sentence level.

Ext-Transformer

Such extractive summarizers utilize a Transformer encoder to learn pairwise interactions (Vaswani et al., 2017) between sentences in a data-driven manner, employing a fully connected priori. In accordance with Liu & Lapata (2019b), an extractor is implemented featuring the same word encoder as the Transformer, succeeded by 12 Transformer-based sentence encoder layers.

HETERSUMGRAPH

The heterogeneous Summarization Graph model delineates relations between sentences via their shared terms, referred to as sentence-word-sentence interactions. This framework encompasses the single-document variant HETERSUMGRAPH (Brody, Alon & Yahav, 2021), as well as its multi-document counterpart, HETERDOCSUMGRAPH. In the multi-document model, node classification is directly employed for sentence selection in the summary, whereas the single-document model utilizes trigram blocking to diminish redundancy.

Our model

The proposed multi-granularity summarization model on a heterogeneous graph establishes relationships between sentences, predicated on both the topics within documents and the shared words among sentences. It integrates dynamic graph attention mechanisms with both adaptive breadth and depth approaches for nuanced learning of node relationships.

Results and analysis

Single-document summarization

The single-document model is assessed using the CNN/DailyMail dataset, reporting overlap metrics of unigram, bigram, and longest common subsequence in relation to R-1, R-2, and R-L scores. Given the constraints of computational resources, a pre-trained contextual encoder was not incorporated into the mode (Devlin et al., 2018). Consequently, comparisons are made solely among models that do not employ BERT.

Results on CNN/DailyMail

Table 1 delineates the model’s performance on the CNN/DailyMail dataset. The initial section of Table 1 features both the LEAD-3 baseline and the ORACLE upper limit, followed by a section that includes other summarization models.

Table 1 Single document results in the CNN/DailyMail dataset.

Model	R-1	R-2	R-L	
LEAD-3 (See, Liu & Manning, 2017)	40.34	17.70	36.57	
ORACLE (Liu & Lapata, 2019b)	52.59	31.24	48.87	
REFRESH (Narayan, Cohen & Lapata, 2018)	40.00	18.20	36.60	
LATENT (Zhang et al., 2018)	41.05	18.77	37.54	
BANDITSUM (Dong et al., 2018)	41.50	18.70	37.60	
NEUSUM (Zhou et al., 2018)	41.59	19.01	37.98	
JECS (Xu & Durrett, 2019)	41.70	18.50	37.90	
LSTM+PN (Zhong et al., 2019a)	41.85	18.93	38.13	
HER W/O POLICY (Luo et al., 2019)	41.70	18.30	37.10	
HER W POLICY (Luo et al., 2019)	42.30	18.90	37.60	
EXT-BILSTM	41.59	19.03	38.04	
EXT-TRANSFORMER	41.33	18.83	37.65	
HETERSUMGRAPH (Wang et al., 2020)	42.95	19.76	39.23	
AREDSUM-CTX (Bi et al., 2020)	43.43	20.44	39.83	
RHGNNSumExt (Chen, 2023)	42.39	19.45	38.85	
Our single-document model	43.14	19.94	39.43	
Our single-document model + Tri-Blocking	43.70	20.19	39.86	
Note:

Performance of our proposed model against recently published summarization systems on the CNN/DailyMail dataset.

In the third section, the proposed heterograph model is presented. Relative to HETERSUMGRAPH, the heterograph records improvements exceeding 0.8/0.4/0.7 on the R-1, R-2, and R-L metrics, respectively. This suggests that the word-sentence-topic-sentence architecture is adept at learning optimal sentence relations through adaptive breadth and depth. Furthermore, the model surpasses the Ext-Transformer based on fully connected relations as well as the one based on sequential structure.

Upon reviewing Table 1, it becomes evident that the proposed model excels over non-BERT-based summary models as denoted in the second block of Table 1. HER (Luo et al., 2019) emerges as a model of comparative efficacy, employing reinforcement learning to recast extractive summarization as a context-bandit problem. Given that both reinforcement learning and triplet chunking serve parallel functions in summarization (Zhou et al., 2018), an additional comparison is made between HER and the proposed model, excluding policy gradients. The model achieves an improvement of 1.42 over R-1 relative to HER when policy gradients are omitted, and 1.40 when incorporating triplet blocks. These metrics suggest that the established graph structure, encompassing sentences and key words, efficaciously amalgamates sentence nodes containing pivotal information, thereby enhancing the efficiency and effectiveness of downstream summarization tasks.

Ablation on CNN/DailyMail

To elucidate the contributions of individual modules, an ablation study is conducted using the CNN/DailyMail dataset. Initially, both the edge weight filtering mechanism and words with low TF-IDF values are removed. Subsequently, any remaining connections between the GAT layer and the adaptive depth layer are eliminated. As a compensatory measure, the updated sentence node is connected to the original sentence node feature in the ensuing formula.

Htt+1=FFN(Depth(Ut←s1)∥Htt).

Moreover, the iteration number for nodes is configured to t = 0, with the subsequent removal of topic node updates and word node updates. Sentence representation Hs1 is directly employed in node classification. Lastly, the BiLSTM layer is excised from both the sentence and topic node initialization procedures.

Table 2 reveals that the elimination of adaptive depth layers results in a decrease in the R-2 metric, while R-1 and R-L metrics remain relatively stable. Given that R-2 in the ROUGE scoring system assesses bigram overlap, it is posited that adaptive depth layers contribute significantly to bigram generation. Consequently, their removal leads to a decline in the R-2 value. Similarly, excising topic nodes adversely affects the model, manifesting more prominently in the decline of the R-1 score. The R-1 metric gauges the concordance between individual words in the generated summary and those in the reference summary. The absence of topic nodes restricts the model’s access to thematic content in the text, impeding the generation of topic-relevant vocabulary. This limitation results in diminished word matching, culminating in a reduction in the R-1 score. Additionally, initializing both topic and sentence nodes with BiLSTM and updating topic nodes corroborate the algorithm’s efficacy. It is conjectured that the quality of summaries generated by extractive models can be enhanced through the inclusion of explicit topic words or phrases. Topic nodes bolster cross-sentence associations, particularly among sentences addressing analogous topics, thereby facilitating the management of cross-sentence dependencies in extensive documents.

Table 2 Ablation studies on CNN/DailyMail test set.

Model	R-1	R-2	R-L	
Our model	43.70	20.19	39.86	
–Filter words	43.39	19.41	39.32	
–Edge ferture	43.21	19.53	39.13	
–Residual connection	42.32	19.23	39.93	
–Topic update	42.77	19.81	39.61	
–Sentence update	42.95	19.86	39.07	
–Word update	42.72	19.92	39.17	
–BiLSTM	42.55	19.54	39.25	
–Adaptive depth layer	43.61	19.32	39.78	
Note:

We removed various modules and explored their influence on our model. A dash ‘–’ means we remove the module from the original model.

Multi-document summarization

Initially, as a baseline metric, the top k sentences from each source document in the dataset are concatenated. Subsequently, additional models employ the output generated from the code and model made public by Fabbri et al. (2019).

Results on multi-news

In the context of multi-document summarization experiments, the model is configured for multi-document analysis, and multiple single documents are concatenated to form an extensive document. The task of multi-document summarization prohibits the use of triple blocking, as the task demands comprehension of several individual documents to generate a comprehensive summary. Results are displayed in Table 3.

Table 3 Multi-document results in Multi-News.

Model	R-1	R-2	R-L	
First-1	25.44	17.30	–	
First-2	35.70	21.60	–	
First-3	40.21	12.13	37.13	
ORACLE	52.32	22.23	47.93	
LexRank (Erkan & Radev, 2004)	41.77	13.81	37.87	
TEXTRANK (Mihalcea & Tarau, 2004)	41.95	13.86	38.07	
MMR (Carbonell & Goldstein, 1998)	44.72	14.92	40.77	
PG (Linmei et al., 2019)	44.55	15.54	40.75	
BOTTOMUP (Gehrmann, Deng & Rush, 2018)	45.27	15.32	41.38	
HI-MAP (Fabbri et al., 2019)	45.21	16.29	41.38	
MGSum-ext (Jin, Wang & Wan, 2020)	45.04	15.98	41.87	
HETERDOCSUMGRAPH (Wang et al., 2020)	46.05	16.35	42.08	
Our multi-document model	46.75	16.47	42.82	
Note:

The answer on Multi-News.

It is observed that the model outperforms prior approaches in multi-document contexts, with particular improvements in various multi-document metrics. The incorporation of topic nodes and adaptive layers contributes positively to the performance of the multi-document summarization model. Concurrently, it is noted that even though document nodes are not directly linked to topic nodes, the designed message-passing mechanism for iterative updates enables topic nodes to convey information to sentence nodes. This information is subsequently transferred from sentence nodes to word nodes, culminating in the updating of document nodes based on the amalgamation of topic and sentence information.

This methodology enables enhanced capture of global semantic information within the text. Word nodes serve as conduits, aiding the flow of information from sentence and topic nodes to document nodes. This connectivity and message-passing technique contribute to an enriched modeling of textual context and relationships, thereby bolstering the model’s capacity to address long-distance dependencies. Additionally, the growing number of documents exacerbates the problem of cross-sentence dependency if sentence nodes are not explicitly connected at the topic level.

Analysis of feature extraction methods

The methodology for extracting features from primary topic nodes and sentence nodes is an essential aspect of the model’s training. Beyond the utilization of BiLSTM, incorporation of BiGRU has also been explored. BiGRU (Cho et al., 2014), similar to LSTM, employs gating mechanisms but usually requires fewer parameters and incurs lower computational overhead. Although BiLSTM generally excels in managing long-range dependencies owing to its specialized gated mechanisms, BiGRU also demonstrates competence in handling such dependencies, albeit possibly necessitating additional time steps.

Initial vector representations for sentences and topics, denoted by Xs∈Rn∗ds and Xt∈Rj∗dt, are acquired via GloVe word embeddings. Following this, BiGRU is applied to glean global features, Gj, while Convolutional Neural Networks (CNN) with diverse kernel sizes are utilized to capture local n-gram features, Lj. The amalgamation of CNN-derived local features and BiGRU-obtained global features results in the sentence node features Xsj=[Lj;Gj] and topic node features Xti=[Li;Gi].

Figure 5 presents the ROUGE score trajectories for both BiLSTM and BiGRU. It is evident that BiLSTM’s enhanced capability in managing long-distance dependencies renders a more favorable impact on the ROUGE-L score. Balancing the trade-offs between training speed and model quality, it is concluded that BiLSTM serves as a more appropriate choice for lightweight models.

Figure 5 The impact of applying different feature extraction methods on ROUGE-L scores on the CNN/DailyMail dataset.

An extractive example

The extracted sentences are shown in Fig. 6, where we have highlighted the theme words of the original text in red. In the summaries generated by the model, these theme words are effectively included, and we have also highlighted them in red in the summary. This indicates that the model can produce sufficiently informative summaries.

Figure 6 An extractive example.

Conclusion

The research introduces a multi-granularity adaptive automatic summarization model predicated on a heterogeneous graph neural network. This approach innovatively incorporates topic semantic units into the heterogeneous graph, thereby enriching the complexity of inter-sentence relations. Attention is also paid to the relative constancy of attention coefficients and the significance of edge features. Adaptive techniques are employed to update nodal characteristics across varying granularities, both in terms of breadth and depth. The proposed model further allows for effortless inclusion of document-level nodes into a single-document framework, thereby simplifying the construction of multi-document summarization models. The designed graph structure—incorporating sentences, topic words, and individual words—can be seamlessly integrated as a module within existing generative models. This integrated module efficiently encodes sentence-level information and channels it into broader linguistic and thematic structures through the graph architecture. Our model, serving as an encoding module, can be applied to other datasets by simply extracting sentences, topics, and words from the dataset and incorporating them into the corresponding graph structure nodes. Performance assessments on the CNN/DailyMail and Multi-News datasets reveal that the model delivers commendable results, particularly when benchmarked against models not leveraging BERT.

Limitations

The model assumes a basic premise that multiple source files share at least one theme. This assumption may limit the applicability of the model in document sets with non-existent or significantly different thematic structures. Future research directions include exploring the complex relationships of entities and events, the sparsity of information, and the density of data in various input fields (such as dialogue summarization, long document summarization) for this model. Future avenues also include the potential for pre-training the model, paralleling other large-scale pre-trained neural summarizers, which would necessitate an additional encoding layer, thereby increasing the model’s complexity. For the purpose of training a BART-based architecture, a GPU endowed with a minimum of 32 GB memory capacity is a requirement. Subsequent endeavors may explore the feasibility of model distillation into more compact configurations without compromising performance efficacy.

Supplemental Information

Supplemental Information 1 Graph attention function.

Click here for additional data file.

Supplemental Information 2 Testing functions.

Click here for additional data file.

Supplemental Information 3 Position encoding function.

Click here for additional data file.

Supplemental Information 4 Encoding function.

Click here for additional data file.

Supplemental Information 5 Data loading and graph structure creation functions.

Click here for additional data file.

Supplemental Information 6 Adaptive depth functions.

Click here for additional data file.

Supplemental Information 7 Graph attention function.

Click here for additional data file.

Supplemental Information 8 Graph attention function.

Click here for additional data file.

Supplemental Information 9 Training functions.

Click here for additional data file.

Supplemental Information 10 Graph model.

Click here for additional data file.

Additional Information and Declarations

Competing Interests

Author Contributions

Data Availability

The authors declare that they have no competing interests.

Wu Su performed the experiments, analyzed the data, performed the computation work, prepared figures and/or tables, and approved the final draft.

Jin Jiang conceived and designed the experiments, prepared figures and/or tables, authored or reviewed drafts of the article, and approved the final draft.

Kaihui Huang performed the experiments, prepared figures and/or tables, authored or reviewed drafts of the article, and approved the final draft.

The following information was supplied regarding data availability:

The code for multi granularity adaptive document summarization based on heterogeneous graph neural networks are available in the Supplemental Files.

The CNN/DailyMail dataset is available at GitHub: https://github.com/abisee/cnn-dailymail.

The Multi News dataset is available at GitHub: https://github.com/Alex-Fabbri/Multi-News.

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
