# Peer review of "Multi-granularity adaptive extractive document summarization with heterogeneous graph neural networks"

_PeerJ Computer Science, doi:10.7717/peerj-cs.1737_

## Round 0.1 · original submission · Major Revisions

The article has values but also has some issues. Please revise the article according to the comments. Then it will be evaluated again.

**Language Note:** The review process has identified that the English language must be improved. PeerJ can provide language editing services - please contact us at copyediting@peerj.com for pricing (be sure to provide your manuscript number and title). Alternatively, you should make your own arrangements to improve the language quality and provide details in your response letter. – PeerJ Staff

Reviewer 1 ·

Basic reporting

1. The paper contains several sentences that are not clear or lack coherence.
2. Errors in grammar and sentence structure.
3. References are not links within the text.
4. Size of the Figures are inconsistently large
5. Mistakes in Numbering under the heading EXPERIMENT
6. In abstract "Ablation studies and model experiments have demonstrated the effectiveness of our model." Write clear results. This doesn't show anything.

Experimental design

1. Please mention the shortcomings or areas where this cannot be implemented.
2. Show error analysis
3. Clarify how these contributions address existing limitations or enhance the state-of-the-art in the field of heterogeneous graphs and inter-sentence relationships.
4.

Validity of the findings

1. Clearly state the significance or implications of the achieved results. How do these results contribute to the broader field of research? What are the potential applications or benefits of the proposed model in real-world scenarios?

2. Mention whether the findings or techniques in the paper have the potential for generalizability to other datasets or domains. This would indicate the broader applicability and relevance of the proposed model beyond the specific dataset mentioned.

·

Basic reporting

The paper targets extractive summarization of textual information, and the authors present an approach based on modeling heterogeneous graphs at three different levels to address the problem.

Overall, the authors provide a viable solution to the defined problem. Yet there are some issues to be addressed concerning the novelty, experiments, and writing quality.

Experimental design

The decision choices within the proposed approach need appropriate justification. For example, why use GloVe embedding, CNN, and BiLSTM in particular for the specific tasks in the overall framework? what are the possible alternatives, and how the decisions are made to select those particular techniques?

The compared methods in Experiments (see Table 1) are largely out of date. More recent methods (published after 2019) should be used to demonstrate whether the proposed work really achieve or surpass the SOTA performance.

Validity of the findings

There is a lack of meaningful discussions to draw some insightful conclusions from the experimental results. For example, what do the results in Table 2 mean to the authors and the research domain? What are the lessons learned from this work? What are the useful hints to future researchers?

Additional comments

The paper can be better motivated. There should be more effective related work discussion to set up the relationship between previous studies and this paper, and also to clearly point out how this work stands out or is distinct from the previous studies. Without such discussion, the novelty of the paper cannot be justified.

---

## Round 0.2 · Minor Revisions

The authors have addressed some issues. However, there are still some typos in the manuscript. Moreover, some details are not described clearly. Please carefully revise the paper accordingly.

Reviewer 1 ·

Basic reporting

1. At some places, indentation at the start of the paragraph is given and at other places its missing, line 238 and 244.
2. Mistake in heading 'graph initializer' at line 162

Experimental design

1. Line 427, 'probing the model’s limitations across various input realms, such as dialogue summarization' - explain model's limitation

Validity of the findings

1. Line 405, 'In comparison to the original text, satisfactory results have been achieved by the model under study.'- What do you mean by satisfactory results?

---

## Round 0.3 · accepted · Accept

Thanks to the authors for their efforts to improve the work. I believe the current version may be accepted.